# Research on Hysteretic Behavior of FRP-Confined Concrete Core-Encased Rebar

**DOI:** 10.3390/polym15122728

**Published:** 2023-06-18

**Authors:** Jingzhou Lu, Tong Mou, Chen Wang, Han Huang, Wenyu Han

**Affiliations:** School of Civil Engineering, Yantai University, Yantai 264005, China; xyc353934154@163.com (C.W.); hhzgq2580@s.ytu.edu.cn (H.H.); hanwenyu@s.ytu.edu.cn (W.H.)

**Keywords:** FCCC-R, hysteretic behavior, cyclic loading, rebar buckling

## Abstract

FRP-confined concrete core-encased rebar (FCCC-R) is a novel composite structure that has recently been proposed to effectively delay the buckling of ordinary rebar and enhance its mechanical properties by utilizing high-strength mortar or concrete and an FRP strip to confine the core. The purpose of this study was to study the hysteretic behavior of FCCC-R specimens under cyclic loading. Different cyclic loading systems were applied to the specimens and the resulting test data were analyzed and compared, in addition to revealing the mechanism of elongation and mechanical properties of the specimens under the different loading systems. Furthermore, finite-element simulation was performed for different FCCC-Rs using the ABAQUS software. The finite-element model was also used for the expansion parameter studies to analyze the effects of different influencing factors, including the different winding layers, winding angles of the GFRP strips, and the rebar-position eccentricity, on the hysteretic properties of FCCC-R. The test result indicates that FCCC-R exhibits superior hysteretic properties in terms of maximum compressive bearing capacity, maximum strain value, fracture stress, and envelope area of the hysteresis loop when compared to ordinary rebar. The hysteretic performance of FCCC-R increases as the slenderness ratio is increased from 10.9 to 24.5 and the constraint diameter is increased from 30 mm to 50 mm, respectively. Under the two cyclic loading systems, the elongation of the FCCC-R specimens is greater than that of ordinary rebar specimens with the same slenderness ratio. For different slenderness ratios, the range of maximum elongation improvement is about 10% to 25%, though there is still a large discrepancy compared to the elongation of ordinary rebar under monotonic tension. Despite the maximum compressive bearing capacity of FCCC-R is improved under cyclic loading, the internal rebars are more prone to buckling. The results of the finite-element simulation are in good agreement with the experimental results. According to the study of expansion parameters, it is found that the hysteretic properties of FCCC-R increase as the number of winding layers (one, three, and five layers) and winding angles (30°, 45°, and 60°) in the GFRP strips increase, while they decrease as the rebar-position eccentricity (0.15, 0.22, and 0.30) increases.

## 1. Introduction

Earthquakes are common natural disasters in nature, and the time and location of their occurrence cannot be accurately predicted by current science and technology. Normally, when a violent earthquake occurs, its tremendous destructive force will cause a devastating blow to the safety of people’s lives and property, and reinforced concrete structures are often subjected to enormous cyclic loading forces. This action effect is often characterized by short duration and a small number of cyclic loads, and significant plastic deformation will take place [1,2], causing great damage to the structure in a short time. On the one hand, When the structure is subjected to a low cyclic load after the protective layer of concrete has been peeled away, it is easy for rebar to fracture at an early stage, which will cause the column to lose its lateral bearing capacity and result in structural collapse [3]; however, on the other hand, rebar buckling is also the primary failure mode of reinforced concrete structures under seismic action, which plays an essential role in the seismic resistance of the structure [4,5,6,7,8]. Therefore, improving the fracture properties and buckling resistance of rebar is beneficial to the bearing capacity and stability of building structures under seismic action.

With the aim of ensuring the safety of building structures under seismic action and giving full play to the properties of rebar, numerous methods have been proposed by researchers to enhance the stability of longitudinal rebar. These include densifying stirrups in the plastic-hinge area of the member to limit the buckling of the longitudinally loaded rebar, which is a widely used method at present [9,10,11]. The appropriate arrangement of transverse rebar significantly improves the seismic performance of reinforced concrete structures but it has always been an issue that makes the binding process of rebar complex and not conducive to the pouring of concrete. In addition, Mitra et al. [12,13] invented an antibuckling structure consisting of a rebar-restraining ring. This research uses a steel pipe to restrain the rebar, which has a fine restraining effect on thinner rebar but is limited to the ordinary reinforced concrete column. FRP has been widely used in construction projects in recent years due to its excellent lightweight, high-strength, corrosion resistance, and ease of cutting properties [14]. Extensive research has been carried out on the application of FRP materials on building structures and reinforced concrete structures [15,16,17,18,19]. As can be seen, FRP materials are widely used in the field of construction engineering due to their unique advantages. Feng et al. [20] inserted a section steel member into an FRP pipe filled with mortar and carried out a monotonic axial compression test. The effects of the cross section of the core steel, slenderness, and FRP fabric layers wrapped at the ends of the specimens were investigated. As a result, these composite components significantly improved the monotonic compressive performance and ductility compared with ordinary section steel members. The monotonic compression ability increased by 44–215% and the ductility increased by up to 877%. Wang et al. [21] proposed a new composite structural FRP-confined concrete core-encased rebar (FCCC-R). Research shows that it can effectively enhance the monotonic compressive capacity of the rebar. The effects of the slenderness ratio, FRP pipe diameter, mortar strength, and rebar-position eccentricity on the monotonic compression capacity of FCCC-R were experimentally studied. An additional parametric study was conducted based on the finite-element model. Lastly, the minimum FRP tube diameter, considering the enhancement effect of monotonic compressive performance, was given, and design curves were proposed for FCCC-R with three types of internal rebar with nominal yield strengths of 400 MPa, 800 MPa, and 1100 MPa. On this basis, Lu et al. [22] also investigated the influence of GFRP strips with different winding layers and angles on the monotonic compression performance of FCCC-R specimens. By performing monotonic compression tests on a series of FCCC-R specimens, it was concluded that their monotonic compression performance increased as the increase in GFRP strips’ winding layers and winding angles and decreased as the slenderness ratio increased. A decrease in the slenderness ratio from 22.73 to 15.45 resulted in an increase in the bearing capacity and ductility of the specimens by 33% and 175%, respectively. Wang et al. [23] studied the cyclic axial compressive capacity of RC columns with FCCC-R. It has a much higher axial load capacity compared to ordinary RC columns. A section superposition was proposed to predict load-deformation behavior. Hu et al. [24] investigated the shear behavior of FCCC-R and explored the impact of different influencing factors on its shear performance. Finally, design equations for the shear strength of FCCC-Rs were proposed. It is worth mentioning that FRP composite pipes have excellent corrosion resistance, so they can be applied to marine structures or saline–alkali areas subject to long-term sulfate attacks to improve their durability [25,26]. For example, the strength retention rate of GFRP bars after 100 years of service in a salt solution with an annual mean temperature of 32 °C will still be higher than 82% [27]. The external GFRP strip of FCCC-R can effectively prevent chloride ions and other chemicals from corroding the internal steel bars and the mortar. However, the mechanical response of steel to cyclic loading differs from monotonic loading due to the presence of the Bauschinger effect, cyclic hardening or softening, and stress relaxation [28,29,30]. Since the rebar in reinforced concrete structures also bears a cyclic load under the action of seismic force, it should be noted that the mechanical performance response of FCCC-R under a monotonic loading test cannot fully reflect its seismic performance under the action of cyclic seismic force. However, most of the existing research results have focused on the monotonic axial compression and shear properties of FCCC-R, as well as the performance of RC columns with FCCC-R. There is almost no research on the hysteretic performance of FCCC-R under cyclic forces. Rebar plays a crucial role in the seismic performance of RC structures. It is necessary to further understand the changes in the seismic performance of rebar under the constraints of mortar and FRP pipes.

In this study, two cyclic loading systems were used for experimental research on FCCC-R with different slenderness ratios and constrained diameters. Further understanding of the impact of external constraint components on the load-bearing capacity, ductility, and buckling of rebar was obtained through experimental results. In addition, an FCCC-R finite-element model was established using ABAQUS software to predict the stress–strain relationship under cyclic loading. An extended parameter study was also conducted based on the established finite-element model, analyzing the effects of the number of winding layers, winding angle of FRP strips, and rebar-position eccentricity on the hysteretic performance of FCCC-R. Both the experimental results and the proposed finite-element model in this paper have certain reference values for scholars and practitioners.

## 2. Materials and Methods

### 2.1. Specimen Design

The steel bars in the FRP-constrained concrete cored rebar in this study were all HRB400 bars of 22 mm diameter. The externally wound FRP strip was a 2 mm thick glass-fiber-reinforced composite (GFRP), and the fiber direction of the GFRP strip was wrapped at an angle of 90° to the radial direction of the specimen. The FCCC-R specimens were formed by inserting rebar into the middle of a GFRP pipe and filling a mortar of C40 strength between the two. Each end of the rebar was set at a length of 100 mm, the outermost 80 mm of which corresponded to the clamp length of the specimen, and the inner 20 mm was the length reserved to ensure that the load was only applied to the rebar. The remaining middle lengths were restrained lengths wrapped by the mortar and the GFRP strip. The configuration of the FCCC-R specimen is shown in Figure 1.

### 2.2. Specimen Preparation

The fabrication of FCCC-R specimens in this study primarily included mold fabrication, mortar pouring, mold removal, maintenance, and wrapping of the GFRP strip. The specific production steps are as follows.

Mold making: the fabrication process used PVC pipe as a mold for pouring mortar. Based on the different restraint diameters and restraint lengths of the FCCC-R specimens, the PVC pipes with different diameters were cut to different lengths. Since the ends of the FCCC-R specimens required an exposed steel length of 100 mm, a PVC pipe of that length was intercepted as the pedestal base of the lower end of the mold. One end of the rebar was wrapped with soft plastic and placed in the base to ensure that it was in the center position, as well as to prevent the mortar from permeating in the base. In addition, to keep the mortar from spilling out, the bottom base and the mold were firmly bonded with tape. 

Mortar filling, remolding, and maintenance: in order to ensure that the rebar was finally in the center during the pouring process, the upper rebar was fixed with a fixed mold and then the mortar was evenly distributed throughout the mold by using the percussion vibrating method. The specimens were shaped after standing for 24 h and then transferred to the maintenance room for 28 days. 

Wrapping of GFRP strips: before wrapping the GFRP strips, the exterior surface of the mortar was wiped with alcohol and the surface was sanded to make it smooth. The fiber strips were then cut according to the different constraint sizes of the FCCC-R specimens. Subsequently, the strips were adhered using a binder, then wrapped at an angle of 90° to the radial direction of the specimen in both fiber directions. The binder was a mixture of epoxy resin and hardener at a ratio of 2:1. Each FCCC-R specimen was glued with 4 layers of fiber strips with a total thickness of 2.8 mm.

### 2.3. Test Materials

In this study, the FRP-constrained concrete cored rebar specimens were all HRB400 steel bars with 22 mm diameter. The basic mechanical properties were measured in accordance with GB/T228.1-2010, Tensile Test of Metal Materials at Room Temperature. The results meet the requirements of GB50011-2016, Seismic Design Code for Buildings. The mortar material was commercially obtained from Qingdao Zhuonengda Construction Technology Co., Ltd. (Qingdao, China). To facilitate the comparison of the elongation of specimens under different loading systems, the results of the rebar material-properties test are listed in Section 3.4. To determine the basic mechanical properties of the mortar used for this study, three sets of cubic specimens were made according to GB/T50081-2019, Standard for Test Methods for Physical and Mechanical Properties of Concrete, which were tested according to the standard test methods after curing. The average strength of mortar-cube specimens was 43.6 MPa. GFRP was made of unidirectional glass-fiber cloth and epoxy-resin glue produced by Haining Anjie Co., Ltd. (Haining, China)The mechanical properties of the GFRP strips were measured in accordance with the test methods specified in GB/T3354-2014, Test Method for Tensile Properties of Oriented Fiber-Reinforced Polymer Matrix Composites; and GB/T3856-2005, Longitudinal and Transverse Shear Test Method for Polymer Matrix Composites for Tensile Mechanical Properties, Longitudinal and Horizontal Shear Tests, respectively. The basic mechanical properties of the GFRP measured by the test are listed in Table 1.

### 2.4. Loading System

The test was conducted using an SDS500 electrohydraulic servo dynamic and static universal testing machine along the radial direction of the specimens for the cyclic loading test; using displacement control loading, the strain rate of the loading rate was 0.00025S^(-1). Ten groups of FCCC-R and ordinary rebar specimens were loaded with constant-amplitude and variable-amplitude mixed loading schemes of two cyclic loading systems, namely the tension–tension and tension–compression equal-amplitude and variable-amplitude cyclic loading systems. The loading systems are shown in Figure 2. Taking the multiple of the yield displacement of the rebar as the unit loading amplitude, the magnitude of each loading amplitude increases in turn, and each one is loaded for two cycles.

The specimens under different cyclic loading systems are listed in Table 2, including 6 groups of FCCC-R specimens and 4 groups of ordinary rebar. Specimens A–F refer to the FCCC-R specimen group, and a–d refer to the ordinary rebar group. The influence of different slenderness ratios of rebar, constraint diameter, and the constraint of composite components on the mechanical properties of rebar under cyclic loading is comprehensively analyzed.

## 3. Results and Analysis

### 3.1. FCCC-R Tension and Compression Cyclic Loading Test Phenomenon

The test was loaded according to the predetermined loading system until the specimens were fractured. Due to the different load amplitude of the FCCC-R specimens with different slenderness ratios, their yield displacements also vary. Therefore, the obtained force-displacement curve cannot accurately reflect the hysteresis performance of material unit strain between specimens with different slenderness ratios and is not conducive to their comparison and analysis. It needed to be converted to a stress–strain curve for analysis with the formula σ = *F*/*A* and ε = Δ*l*/*l*. 

Under cyclic loading, the test phenomena of various types of FCCC-R specimens are basically identical. At the initial stage of loading, the deformation of the specimen is mainly elastic. As the loading proceeds, because of the existence of the Bauschinger effect, the specimen soon undergoes plastic deformation during the reverse loading process, and this phenomenon becomes more obvious with an increasing displacement amplitude. The reason for this is that the fiber of the GFRP strip winds circumferentially, and only the bonding force between the fibers produces tension in the radial direction. After the buckling of FCCC-R, the transverse deflection of the specimen increases rapidly and the crack deepens gradually. Finally, with the cracking and destruction of the external restraint components, their utility is greatly reduced and, due to the lack of their support, the damage to the core rebar is aggravated resulting in a large amount of plastic deformation, which causes the specimen to be pulled off. Although the internal filled mortar is crushed and cracked before the fracture of the specimen, the mortar can still maintain a certain shape due to the binding effect of the GFRP strip and it still has a certain binding capacity. It is worth noting that when the rebar is pulled off, there is no “necking” phenomenon as in the monotonous stretching of ordinary rebar. The fracture section of the core rebar of FCCC-R is basically planar. The fracture section of the ordinary rebar is inclined. The fracture morphologies of both are shown in Figure 3. This phenomenon is caused by the “defect” of the material itself. Under repeated loads, some tiny cracks appear at the defect. From that point, the continuous development and deterioration of the microcracks lead to the weakening of the specimen section and the stress concentration at the crack root, and the plastic deformation is limited, leading to the brittle fracture of the rebar after a certain number of cycles.

### 3.2. Tension–Compression Cyclic Loading Test Results

According to the test results shown in Figure 4, the hysteretic performance of the FCCC-R specimens is significantly better than that of ordinary rebar, which is primarily reflected in the fact that the FCCC-R specimens are able to load more cycles than the ordinary rebar with the same slenderness ratio and has a larger hysteresis envelope area, and the maximum bearing capacity in the compression stage is higher than that of ordinary rebar. At the same time, the hysteretic curve of specimens with a small slenderness ratio and large constraint area is plumper and more symmetrical. The results of this study show that under the same constraint diameter, the maximum compression capacities of the FCCC-R specimens with slenderness ratios of 10.9, 15.5, 20, and 25.5 are 38.7%, 31.1%, 25.0%, and 23.8% higher than those of ordinary rebar, respectively; moreover, the maximum tensile strain before fracture increased by 32.9%, 30.0%, 30.1%, and 20.2%, respectively, and the fracture stress decreased by 42.8%, 1.9%, 11.4%, and 24.8%, respectively. The reduction in the fracture stress of the FCCC-R specimens means that their ductility at fracture is better than that of ordinary rebar [31]. The use of the FRP pipe filled with mortar plays a crucial role in restraining the rebar and fully restrains the premature occurrence of rebar buckling, while also limiting the excessive development of lateral deflections of rebar after buckling. Due to the rapid development of lateral deflections, the root of the crack will propagate and intensify the damage to the rebar, which is more likely to cause brittle fractures to it under stress concentration during reverse tensile loading. In addition, the envelope area of the hysteresis loop increases by 77.8%, 60.9%, 50.4%, and 34.2%, respectively. The comparison of the results is shown in Figure 5. The reason for these phenomena is that the damage to ordinary rebar accumulates faster than that of FCCC-R and the plastic deformation develops more rapidly.

Furthermore, in the process of reciprocating loading, the compression peak stress of members with low slenderness ratios decreases more slowly after buckling, and the rebar damage accumulation is relatively slow. On the contrary, the reduction of compressive peak stress is accelerated, and the damage accumulation is more rapid. To sum up, FCCC-R has better seismic energy-consumption capacity than ordinary rebar. The main mechanical property parameters of FCCC-R cyclic loading with the same constraint diameter in this study are listed in Table 3.

According to the comparison of skeleton curves, with the increase in the slenderness ratio, there is no obvious change on the tensile side of the skeleton curve except for the specimen with a loading length of 240 mm. The influence on the cyclic loading of tension–compression is mainly reflected in the compression performance. There is a more obvious increase in peak compressive stress. This is because the mortar and GFRP strips have a better restraint effect when the lateral deflection of the rebar occurs, thus, inhibiting the development of the transverse deflection. Moreover, when the rebar does not generate a large transverse deformation, the compression results in an increase in the cross-sectional area, and the friction resistance between the rebar and mortar also increases the compression bearing capacity to a certain extent [30]. When the specimen is in tension, it is more the friction between the mortar and rebar that acts on the restraint of the rebar. The specimen with a loading length of 240 mm is different from other specimens in the result of significantly increasing the maximum bearing capacity at the tensile side. In this case, it is considered that its buckling occurs late, which clearly demonstrates the cyclic strengthening phenomenon, increasing the peak stress of the test piece with the amplitude of each stage of cyclic loading before buckling.

The influence of the constraint diameter on the hysteretic performance of FCCC-R is also studied here. Three groups of samples with restraint diameters of 30, 40, and 50 mm were used for cyclic loading tests on specimens with a loading length of 440 mm. In agreement with the above results, the constraint diameter had a small impact on the tensile bearing capacity and a large impact on the compressive bearing capacity. With the increase in the restraint diameter, the maximum compressive capacity and the envelope area of the hysteretic ring for each cycle of cyclic loading also increased. The maximum compression bearing capacity of specimens E, C, and F was 7.7%, 30.0%, and 43.7% higher than that of ordinary rebar. The shape of the FCCC-R skeleton curve of each constraint diameter was basically similar, but with the increase in the constraint diameter, the buckling phenomenon appeared later. The FCCC-R specimens with constrained diameters of 30 mm, 40 mm, and 50 mm buckled in the first cyclic compression stage of the second, third, and fourth yield displacement, and the final maximum strain before fracture was 0.025, 0.029, and 0.027. The results are listed in Table 3. In terms of the envelope area of the hysteresis loop and the maximum strain value, the improvement effect of specimen F was not as good as that of specimen C, which does not reflect the regular pattern of the hysteretic performance of FCCC-R increasing with the constraint diameter. Through analysis of its causes, this phenomenon is thought to be caused by the insufficient vibration of mortar in the process of making the specimen or the defects of the rebar itself, which is a premature fracture of the specimen in the process of loading. The premature fracture of specimen F results in a reduction in the number of loading cycles. However, under the same number of loading cycles, the area of the hysteresis loop of specimen F is larger than that of test piece C. Therefore, the FCCC-R specimens with a larger constraint diameter still have better hysteresis performance. When the constraint diameter is 30 mm, the difference between the skeleton curve and the ordinary rebar is small, indicating that the smaller constraint diameter has a very limited effect on improving the hysteretic performance of the rebar. It is, thus, concluded that with the increasing rebar slenderness ratio, only by using the appropriate restraint diameter can an FCCC-R specimen with fine hysteretic performance be obtained. The value relationship of the best constraint diameter under a certain slenderness ratio of rebar needs to be further explored.

### 3.3. Tension–Tension Cyclic Loading Test Results

The test phenomenon of FCCC-R specimens under tension–tension cyclic loading is roughly the same as that under tension–compression cyclic loading, which is not described in this section. However, the location of the rebar fracture occurs at the end of the restraint length. The fracture position of the ordinary rebar is still in the middle of the specimen. Six groups of specimens with slenderness ratios of 10.9, 15.5, and 20, including ordinary rebar and FCCC-R specimens with a constraint diameter of 40 mm, were loaded under tension–tension cyclic loading until their fracture. The test results are shown in Figure 6. 

The hysteretic performance of FCCC-R specimens is significantly improved compared with that of ordinary rebar, which is the same as under tension–compression cyclic loading. Furthermore, their hysteretic properties also increase with the decrease in slenderness ratio. As the slenderness ratio increases under the same constraint conditions, the envelope area of the hysteresis loop of the FCCC-R specimens is 52.0%, 57.1%, and 12.5% higher than that of the ordinary rebar; the maximum tensile strain before fracture increases by 23.7%, 25.0%, and 0% and the maximum compression bearing capacity increases by 19.0%, 18.2%, and 28.3%. The comparison of the results is shown in Figure 7. Moreover, the maximum compressive bearing capacity of the FCCC-R specimens is higher than the yield strength of the ordinary rebar. The restraint components are more conducive to limiting the lateral deflection of FCCC-R with larger slenderness ratios. Among the three, FCCC-R specimens with a slenderness ratio of 20 have a greater improvement in the maximum compressive bearing capacity of the rebar. Under the same slenderness ratio, the maximum tensile bearing capacity of the FCCC-R specimens is still not significantly higher than that of ordinary rebar. Another reason is that the external constraint component plays a very small role in the tensile process. The law is the same as the one illustrated in Section 3.2. Table 4 shows the major hysteretic capacity parameters for both specimen types under cyclic tension–tension loading.

### 3.4. Comparison of Elongation under Different Loading Systems

Ductility refers to the ability of the structure or member to resist plastic deformation with no obvious reduction in bearing capacity and refers to the deformation ability of the material, component, and structure after yielding. Elongation is one of the main indicators reflecting the deformation capacity of rebar.

In this study, the monotonic tensile test of ordinary rebar was compared with the elongation of ordinary rebar and some FCCC-R specimens under two cyclic loading systems. The main mechanical properties, and the corresponding elongation of ordinary rebar under monotonic tension, were taken from the test data obtained in the material-property test and directly displayed through the computer-control system connected to the testing machine. The total length of the rebar specimen for the material property test was 500 mm, the loading length was 350 mm, and the loading rate was 0.00025S-1. The test results are listed in Table 5. The specific values are shown in Table 6.

By comparing the elongation of different specimens under different loading systems, in general, the elongation of rebar under cyclic loading is significantly different from that under monotonic loading; the former is significantly smaller than the latter. Based on the comparison of the maximum elongation of the specimens shown in Figure 8, under two different cycling systems, the elongation of rebar under the tension–tension cyclic loading is greater than that under the tension–compression cycling loading. Compared with tension–compression, under tension–compression cyclic loading, the maximum elongation of FCCC-R specimens with slenderness ratios of 10.9, 15.5, and 20 increased by 62.1%, 55.2%, and 27.6%, respectively. The rebar was enhanced by 65.2%, 56.5%, and 47.8%.

There is also a difference in the stress–strain relationship of the specimens under the two cyclic loading systems. This is because the elongation and the stress state of the rebar are related to the cyclic loading history of the rebar [27,28], which indicates that the damage accumulation of the tension–compression system is more severe than that of the tension–tension system. However, regardless of the system, the elongation of FCCC-R specimens under the same loading length is greater than that of ordinary rebar. It can be seen that its ductility is better than that of ordinary rebar and the external constraint improves the ductility of the rebar. However, there is still a gap between the elongation of FCCC-R specimens under cyclic loading systems and that of rebar specimens under monotonic tension. The fracture elongation of FCCC-R specimens under tension–tension cyclic loading can reach approximately 20~30% of the elongation at a maximum force under the monotonic loading of ordinary rebar. This value is about 15~20% under tension–compression cyclic loading. In other words, the existence of external restraint components improves the ductility of rebar under cyclic loading but there is still a big gap between the elongation of rebar under monotonic loading.

### 3.5. Comparison of Mechanical Properties under Different Loading Systems

The objective of this section is to compare and analyze the maximum compressive bearing capacity and envelope area of the hysteresis loop of specimens under two different cyclic loading systems. The results are shown in Figure 9.

Due to the more obvious cyclic hardening effect of steel, the maximum compressive bearing capacity of the specimens under tension–compression cyclic loading is greater than that under tension–tension cyclic loading. The maximum compressive bearing capacity of specimens A–C and a–c increased by 21.7%, 17.4%, 10.1%, 5.1%, 5.2%, and 12.7%, respectively. In contrast, the envelope area of the hysteresis loop was relatively larger under the tension–tension system. Except for specimen C, which decreased by 10.5%, the other five groups of specimens increased by 21.2%, 40.1%, 41.8%, 42.8%, and 18.2%. The reason for this phenomenon is that the tension–tension system has a slower accumulation of damage to the rebar, so more cycles can be loaded.

The basic mechanical properties of specimens C and c under tension–compression cyclic loading are compared with the monotonic compression performance in [22]. From Figure 10, it can be seen that the maximum compressive bearing capacity of specimen C under tension–compression cyclic loading increased by 9.9% compared to monotonic loading, from 473 MPa to 520 MPa. In addition, the rebars are more prone to buckling under cyclic loading. The buckling strain of the FCCC-R is 0.0061 under monotonic loading, while under cyclic loading, this value is only 0.0053. Similarly, the maximum compressive bearing capacity of the ordinary rebar under tension–compression cyclic loading increased from 362.1 MPa under monotonic loading to 415 MPa, and the buckling strain decreased from 0.0019 to 0.0017. From a comparative perspective, whether under monotonic compression or cyclic loading, FCCC-R has a significant effect on improving the maximum compressive bearing capacity of rebars and delaying their buckling. Therefore, FCCC-R has a good effect on fully utilizing the mechanical properties of internal rebars.

## 4. Finite-Element Simulation and Parametric Analysis

### 4.1. Material Constitutive Model

In this study, the finite-element numerical simulation software ABAQUS was used to simulate the hysteretic curve of FCCC-R specimens under tension–compression cyclic loading. We divided the FCCC-R model into three parts—steel, mortar, and a GFRP tube—to establish a finite-element model, in which the mortar and core rebar models were established using solid elements. Due to the low thickness of the GFRP strip compared to the constrained component size, the shell element was used to establish the GFRP tube model. The element type used for the solid element was C3D8R, whereas S4R was used for the shell element. All elements were approximately 6 mm in size. All components were assembled and grids were divided according to the different types of FCCC-R forms. The FCCC-R finite-element model is shown in Figure 11. After observing that the FCCC-R specimen had been damaged after the test, it was found that there was no obvious slippage in the interface between the mortar and the rebar, nor in the interface between the mortar and the GFRP strip. For this reason, in ABAQUS, the “tie” function was used to bind the three parts into pairs. Furthermore, in order to trigger the specimens’ instability during cyclic loading, the shape of the first-order buckling mode with a loading length of 1/1000 in the buckling analysis results was introduced into the finite-element model as an initial geometric defect [22].

### 4.2. Material Constitutive Model

The concrete damage plasticity (CDP) model included in the ABAQUS material library was used to define the three-dimensional constitutive relationship of mortar. The CDP model can be used to simulate the mechanical behavior of concrete structures and composite structures under reciprocating loads and is widely used in seismic analysis. Its main parameters, and the compressive and tensile constitutive relationships, are described in [21]. In the experiment, the filling mortar was subjected to low circumferential confinement from the external composite pipe. Therefore, the influence of the plastic-strain law of concrete on the constraint effect was not considered in this simulation. For the GFRP materials, the mechanical behavior of GFRP was divided into elastic and damage destruction stages. Due to the anisotropic elasticity of GFRP, the model used engineering constants to define the tensile modulus of elasticity, shear modulus, and Poisson’s ratio in three directions. The Hashin damage criterion has been widely used in progressive damage analysis of composite materials and in defining the failure mode of GFRP strips during the damage stage [32,33]. The main parameters in the Hashin damage criterion were taken based on the material property test results, which are shown in Table 1. The Chaboche [34] steel-plastic constitutive model was used to simulate the stress–strain curve of the rebar under cyclic loading. The model assumes von Mises yield criterion and an associative flow rule is assumed. The characteristics of a material are defined by a combined isotropic/kinematic hardening model. Figure 12 is a diagram of the isotropic/kinematic hardening model. This model can effectively simulate the Bauschinger effect and the cyclic hardening or softening of rebar during cyclic loading. The constitutive model parameters of the rebar under cyclic loading were obtained by fitting experimental data. The specific method is shown in [2,35,36,37]. In this study, a total of three sets of backstresses were taken. When setting rebar material properties in ABAQUS, the “combine” function was used to set the material strengthening model; then, the fitted constitutive model parameter values in Table 7 were input.

### 4.3. Comparison of Test and Simulation Results

The objective of this section was to simulate the hysteretic curve of FCCC-R specimens subjected to tension–compression cyclic loading. At the same time, a unit loading amplitude of twice the yield displacement loading system was introduced to conduct a finite-element simulation on specimen C and we labeled this group of samples as specimen G. The simulation results were compared with test data to further verify the accuracy of cyclic strengthening parameters. When the FCCC-R specimens were loaded to the eighth yield displacement, their hysteretic performance was greatly reduced, which also saved model calculation time. Therefore, this study simulates hysteretic curves of the first-eight yield displacements of FCCC-R specimens A–F. For specimen G, it took the same loading time as specimen A–F models for simulation. The simulation and test results of a total of seven groups of specimens were compared, as shown in Figure 13. It can be seen that the simulation results reflect the cyclic hardening and softening, unloading stiffness, peak load, and buckling phenomenon of the rebar under cyclic loading, all of which are in good agreement with the test results. Table 8 summarizes the peak load and hysteresis-loop envelope area obtained by finite-element calculation and compares them with the test results. As can be seen, the discrepancy between the calculated value and the test value is approximately 10%, and only the difference between the calculated value of the hysteresis-loop envelope area of individual specimens and the test value is more than 20%. In general, the simulation and test results agree well, indicating that the constitutive relationship and strengthening parameters used in this study can accurately simulate the stress–strain curve of FCCC-R specimens under tension–compression cyclic loading.

### 4.4. Parametric Study and Analysis

In order to further study the influencing factors on the hysteretic performance of FCCC-R specimens to compensate for the shortcomings of incomplete and superficial testing in this study and based on the previous specimen C finite-element model, we carried out an expanded parametric study. The number of winding layers *L* and winding angles *J* of the GFRP strips, as well as the rebar-position eccentricity *e/R*, were used as analysis parameters for this study. The number of winding layers refers to the total number of layers of GFRP strips wrapped on the outside of the high-strength mortar. The winding angle is the included angle between the fiber direction of the GFRP strips and the surrounding direction of the FCCC-R specimens. The rebar-position eccentricity refers to the ratio of the distance from the center of the rebar section to the center of the GFRP tube to the constraint radius. The value ranges of the parameters are listed in Table 9. For purposes of comparison and analysis, the skeleton curves of the FCCC-R models with different influence parameters are displayed to roughly reflect their hysteretic performance. The simulation results are shown in Figure 14. 

Based on the simulation results, it is concluded that the hysteretic performance of FCCC-R increases to a certain extent with the increase in the number of winding layers and angles of GFRP strips. This is due to the fact that when 0° winding is used, the GFRP strips of the FCCC-R specimens are only affected by the adhesive force of the fibers. However, the tensile bearing capacity of the fibers is greater than the bonding force between them; therefore, as the winding angle increases, the tensile properties of the fibers begin to play an increasingly important role, continuously improving the hysteretic performance of FCCC-R. Similarly, as the number of winding layers on the GFRP strips increases, the superposition of the effectiveness of multilayer GFRP strips also enhances the constraint effect of restraint components. This leads to an improvement in the constraint effect of external components on rebar. As the constraint effect of external components increases, rebar can further exert its mechanical properties, resulting in an increase in peak stress in compression and tension at each cycle of loading. There are some similarities between this enhancement mechanism and increasing the diameter of the external constraint to improve the hysteretic performance of rebar. However, due to the much larger tensile bearing capacity in the fiber direction compared to mortar, on the tensile side of the rebar, GFRP strips have a greater effect on the restraint of the rebar.

Contrary to the above conclusions, the hysteretic behavior of FCCC-R specimens decreases continuously with the increase in rebar-position eccentricity. The reason for this is that the existence of eccentricity leads to the existence of a weak part on one side of the restraint component, which weakens the confined effectiveness of the eccentric side in terms of restraining the rebar capacity when subjected to cyclic loading, resulting in a reduction in the hysteretic performance of the FCCC-R specimens. When the eccentricity reaches 0.22, the hysteretic properties of the rebar will be clearly reduced.

## 5. Conclusions

This study examined the hysteretic behavior of FRP-confined concrete core-encased rebar (FCCC-R) under cyclic loading. The main parameters for evaluating the hysteretic properties of FCCC-R specimens are the loading lengths and constraint diameters. Additionally, the study investigated the effects of different loading systems on these properties. The elongation and mechanical properties of specimens under different cyclic loading systems were compared with those of ordinary rebar under monotonic tension. Finite-element simulation and corresponding expansion parameter analysis were also carried out. The following conclusions are drawn from the study results:

(i) Compared to ordinary rebar, FCCC-R specimens exhibit superior hysteretic performance, as evidenced by their higher compression peak load and fracture strain, increased number of load cycles, larger hysteresis envelope area, and lower fracture stress. Additionally, the buckling phenomenon of FCCC-R specimens during loading is delayed compared to that of ordinary rebar due to the confinement provided by the high-strength mortar and GFRP strips;

(ii) Under the same slenderness ratio, the hysteretic performance of FCCC-R was improved to varying degrees with the increase in the constraint diameter. However, the hysteretic behavior of the specimens with a small constraint diameter was not significantly different from that of the ordinary rebar. Therefore, with the increase in slenderness ratio, it was necessary to appropriately increase the constraint diameter in order to achieve a significant improvement;

(iii) The hysteretic properties of the specimens under tension–compression cyclic loading were consistent with those under tension–tension cyclic loading. The hysteretic properties of the specimens decreased with the increase in loading length. However, due to the small cumulative damage of rebar under tension–tension cyclic loading, its hysteretic performance was better than that under tension–compression cyclic loading;

(iv) Through comparison of the elongation of specimens under different loading systems, it was observed that the FCCC-R specimens exhibited greater elongation under tension–tension cyclic loading than those under tension–compression cyclic loading. In addition, under any cyclic loading system, the fracture elongation of the FCCC-R specimens was greater than that of ordinary rebar but still lower than the elongation of the maximum bearing capacity of ordinary rebar under monotonic tension. Under cyclic loading, specimen buckling occurred earlier but FCCC-R effectively delayed the phenomenon of rebar buckling. Moreover, under tension–compression cyclic loading, the maximum compressive bearing capacity of the FCCC-R specimens was improved compared to monotonic compressive loading;

(v) Finite-element models of different FCCC-Rs were established by using ABAQUS software for numerical analysis. The model was validated through comparison with experimental results. Except for a small amount of data with a discrepancy of 20%, the data discrepancy was approximately 10%. Therefore, the numerical simulation results agreed well with the experimental results. Based on these results, expanded parametric studies were also carried out to analyze the effects of different winding layers, winding angles of GFRP strips, and rebar-position eccentricity on the hysteretic properties of FCCC-R. It was concluded that the hysteretic performance of FCCC-R increased with the increase in the number of GFRP winding layers and winding angles. The increase in rebar-position eccentricity continuously reduced the hysteretic performance of FCCC-R;

(vi) According to the research in this article, when designing FCCC-R, it is advisable to adopt a larger wrapping angle of GFRP strips when the concrete expansion is small. The constraint diameter of the constraint component cannot be too small. In this article, it is recommended that the minimum area ratio of rebar to constrained components be 0.54. In addition, reducing the rebar-position eccentricity as much as possible is conducive to improving the ground hysteretic performance of FCCC-R.

## Figures and Tables

**Figure 1 polymers-15-02728-f001:**
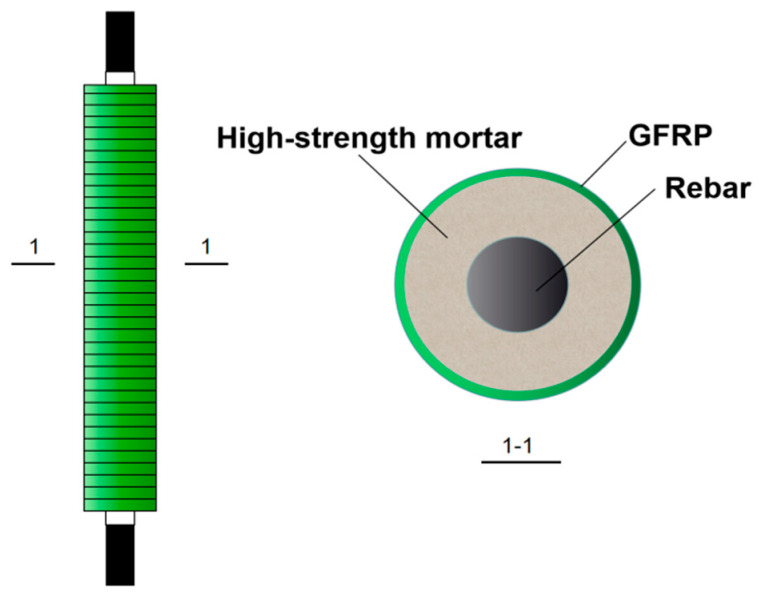
Configuration of FCCC-R specimens.

**Figure 2 polymers-15-02728-f002:**
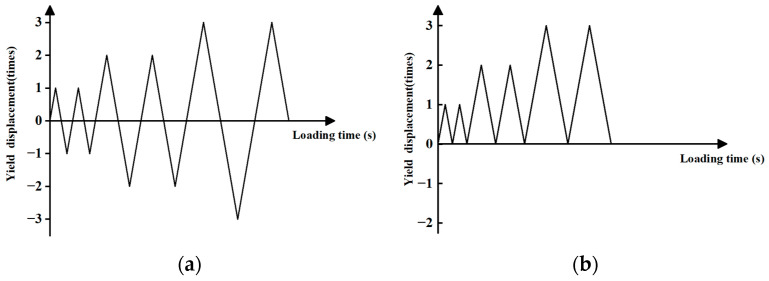
Schematic diagram of cyclic loading systems. (**a**) Tension–compression cyclic loading system; (**b**) Tension–tension cyclic loading system.

**Figure 3 polymers-15-02728-f003:**
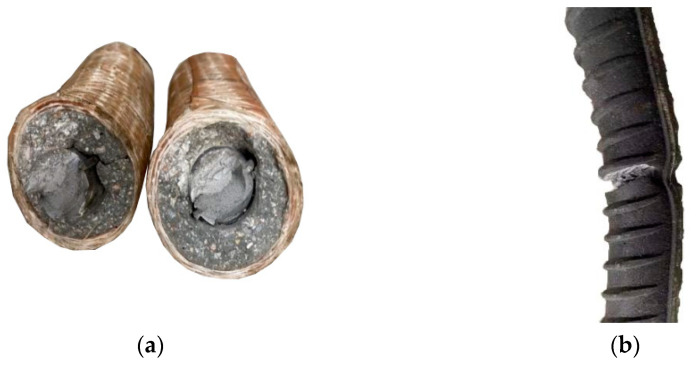
Fracture morphology of specimens. (**a**) Fracture morphology of FCCC-R specimen; (**b**) Fracture morphology of ordinary rebar specimen.

**Figure 4 polymers-15-02728-f004:**
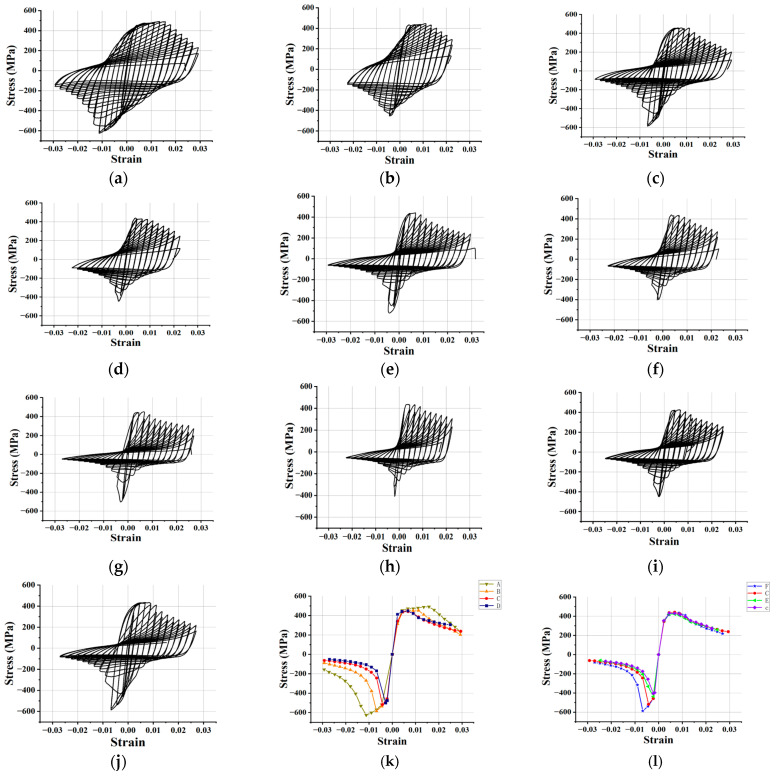
Hysteresis curves of tension–compression cyclic loading. (**a**) Specimen A; (**b**) Specimen a; (**c**) Specimen B; (**d**) Specimen b; (**e**) Specimen C; (**f**) Specimen c; (**g**) Specimen D; (**h**) Specimen d; (**i**) Specimen E; (**j**) Specimen F; (**k**) Comparison of skeleton curves of specimens with different slenderness ratios; (**l**) Comparison of skeleton curves of specimens with different constraint diameters.

**Figure 5 polymers-15-02728-f005:**
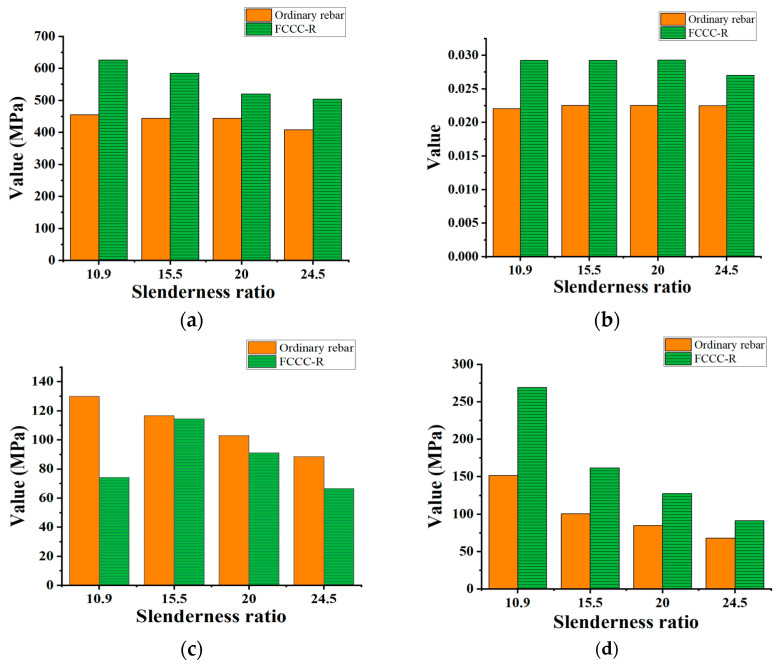
Comparison of basic hysteretic properties between FCCC-R and ordinary rebar under tension–compression cyclic loading. (**a**) Maximum compressive bearing stress; (**b**) Maximum strain before fracture; (**c**) Fracture stress; (**d**) Envelope area of the hysteresis loop.

**Figure 6 polymers-15-02728-f006:**
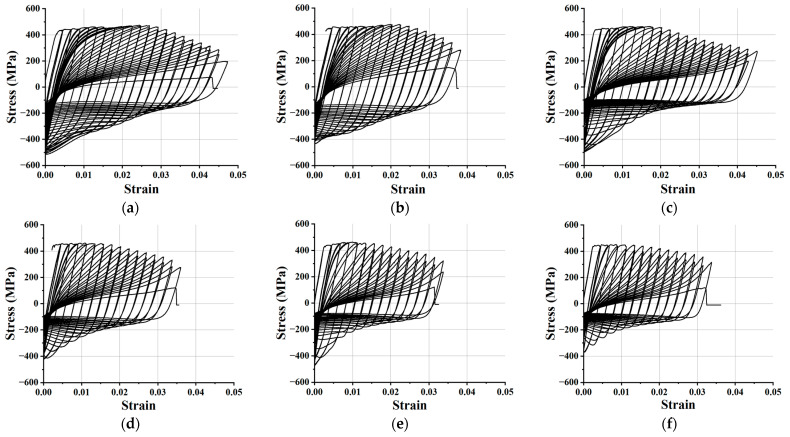
Hysteresis curves of tension–tension cyclic loading. (**a**) Specimen A; (**b**) Specimen a; (**c**) Specimen B; (**d**) Specimen b; (**e**) Specimen C; (**f**) Specimen c.

**Figure 7 polymers-15-02728-f007:**
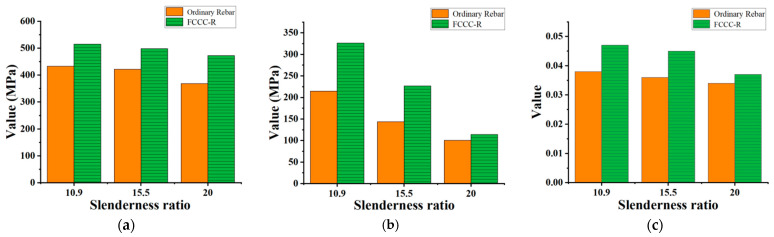
Comparison of basic hysteretic properties between FCCC-R and ordinary rebar under tension–tension cyclic loading. (**a**) Maximum compressive bearing stress; (**b**) Envelope area of hysteresis loop; (**c**) Maximum strain before fracture.

**Figure 8 polymers-15-02728-f008:**
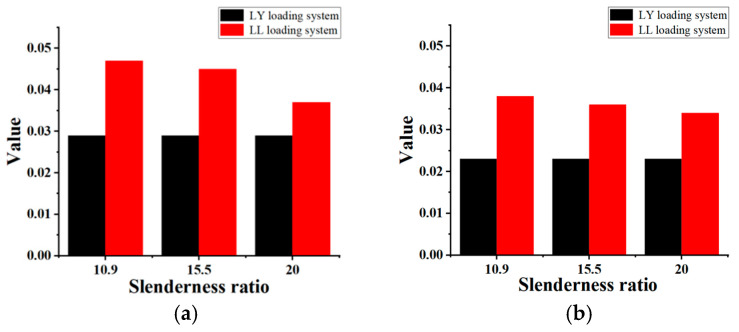
Comparison of maximum elongation of specimens under two loading systems. (**a**) FCCC-R; (**b**) Ordinary rebar.

**Figure 9 polymers-15-02728-f009:**
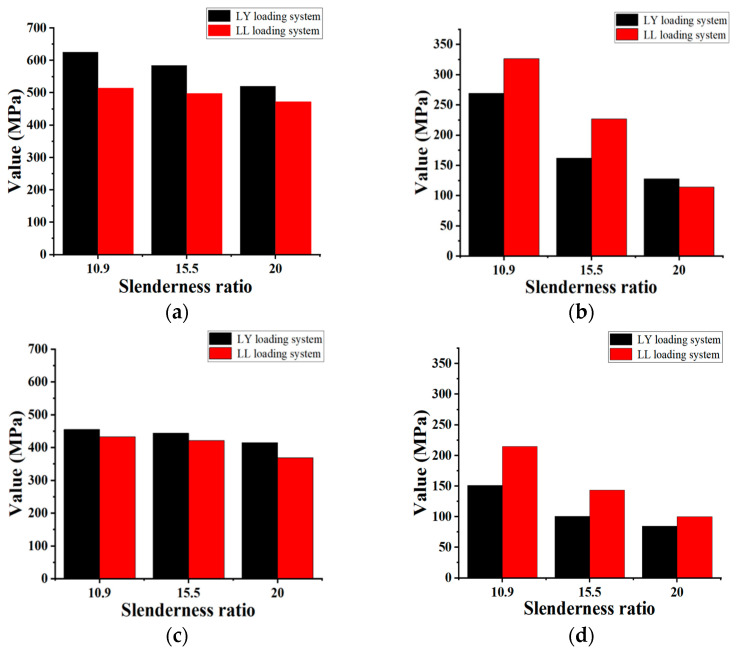
Comparison of mechanical properties of specimens under two loading systems. (**a**) Maximum compressive bearing capacity of FCCC-R; (**b**) Envelope area of the hysteresis loop of FCCC-R; (**c**) Maximum compressive bearing capacity of ordinary rebar; (**d**) Envelope area of the hysteresis loop of ordinary rebar.

**Figure 10 polymers-15-02728-f010:**
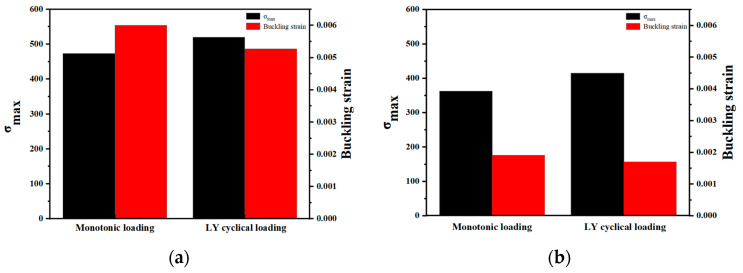
Comparison of mechanical properties between monotonic loading and cyclic loading of FCCC-R. (**a**) FCCC-R; (**b**) Ordinary rebar.

**Figure 11 polymers-15-02728-f011:**
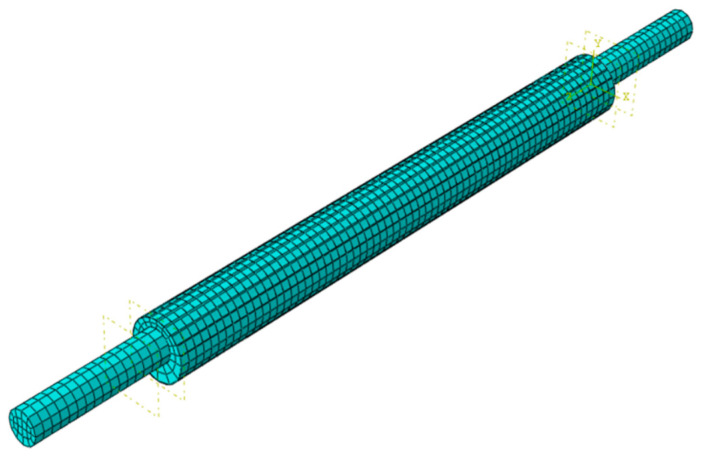
The finite-element model of FCCC-R.

**Figure 12 polymers-15-02728-f012:**
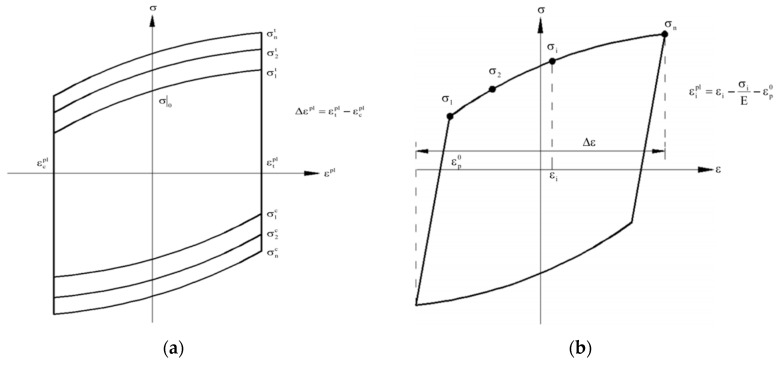
Rebar finite-element model parameter calibration [2]. (**a**) Isotropic hardening component; (**b**) Kinematic hardening component.

**Figure 13 polymers-15-02728-f013:**
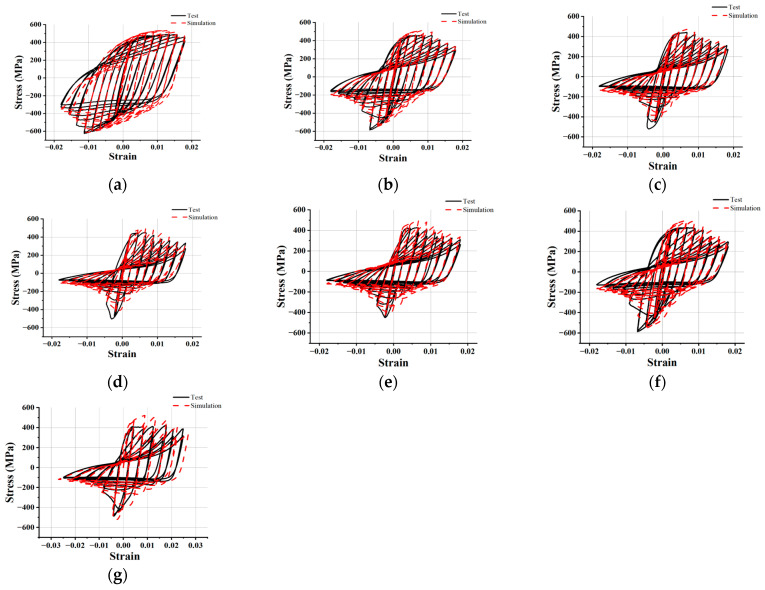
Test and simulation hysteretic curve comparison for FCCC-R specimens. (**a**) Specimen A; (**b**) Specimen B; (**c**) Specimen C; (**d**) Specimen D; (**e**) Specimen E; (**f**) Specimen F; (**g**) Specimen G.

**Figure 14 polymers-15-02728-f014:**
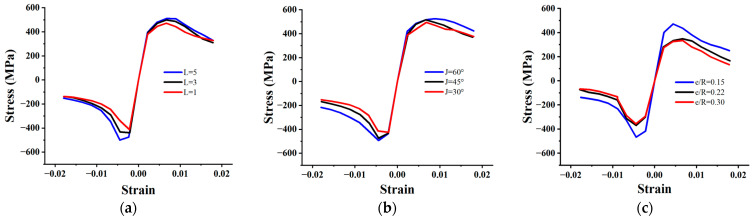
Skeleton curves of FCCC-R with different influence parameters. (**a**) Winding layers; (**b**) Winding angle; (**c**) Rebar-position eccentricity.

**Table 1 polymers-15-02728-t001:** Basic mechanical properties of GFRP strips.

Longitudinal Tensile Strength (MPa)	Transverse Tensile Strength (MPa)	Shear Strength (MPa)	Longitudinal Elastic Modulus (GPa)	Transverse Elastic Modulus (GPa)	Shear Modulus (GPa)	Poisson’s Ratio
760.2	55.1	217.1	45.1	2.7	14.9	0.23

**Table 2 polymers-15-02728-t002:** Main parameters of cyclic loading specimens.

Specimen Number	Loading Length (mm)	Confined Diameter (mm)	Slenderness Ratio	Unit Yield Displacement (mm)	Loading Speed (mm/min)
A	240	40	10.9	0.54	3.6
B	340	40	15.5	0.765	5.1
C	440	40	20	0.99	6.6
D	540	40	24.5	1.215	8.1
E	440	30	20	0.99	6.6
F	440	50	20	0.99	6.6
a	240	0	10.9	0.54	3.6
b	340	0	15.5	0.765	5.1
c	440	0	20	0.99	6.6
d	540	0	24.5	1.215	8.1

**Table 3 polymers-15-02728-t003:** Tension–compression cyclic loading test results.

Specimen ID	*σ*_max_ (MPa)	*σ*_max,t_ (MPa)	*α*	*σ*_max_/*σ*_d_	Maximum Strain Value	Hysteresis Envelope Area (MPa)	Fracture Stress (MPa)
A	626.41	500.32	1.38	1.46	0.029	269.08	74.23
B	584.90	451.25	1.21	1.36	0.029	161.83	114.31
C	520.19	441.08	1.25	1.21	0.029	127.49	91.12
D	503.95	446.40	1.23	1.17	0.027	91.20	66.61
E	446.83	426.74	1.07	1.04	0.025	92.03	71.38
F	582.40	425.90	1.40	1.36	0.027	121.85	51.34
a	454.80	440.82	-	1.06	0.022	151.37	129.83
b	443.39	434.92	-	1.03	0.023	100.58	116.58
c	415.00	434.03	-	0.97	0.023	84.79	102.89
d	408.20	431.23	-	0.95	0.022	67.94	88.52

σ_max_—the maximum compressive bearing stress; σ_max,t_—the maximum tensile stress; α—the ratio of the maximum bearing capacity of the FCCC-R specimens to the ordinary rebar with the same slenderness ratio; σ_d_—the stress at the yield point of the reinforcement under monotonic tension.

**Table 4 polymers-15-02728-t004:** Tension–tension cyclic loading test results.

Specimen ID	*σ*_max_ (MPa)	*σ*_max,t_ (MPa)	*σ*_max_/*σ*_d_	Maximum Strain Value	Hysteresis Envelope Area (MPa)	Fracture Stress (MPa)
A	514.6	473.0	1.20	0.047	326.31	71.46
B	498.1	464.5	1.16	0.045	226.80	192.40
C	472.4	453.1	1.10	0.037	114.04	121.07
a	432.5	473.3	1.01	0.038	214.74	130.46
b	421.3	456.2	0.98	0.036	143.72	119.59
c	368.2	448.6	0.86	0.034	100.21	120.74

**Table 5 polymers-15-02728-t005:** Experimental results of monotonic tensile rebar.

Elastic Modulus (MPa)	Yield Strength (MPa)	Tensile Strength (MPa)	Maximum Stress Elongation (%)	Elongation at Break (%)
190.2	429.0	557.5	13.7	22.8

**Table 6 polymers-15-02728-t006:** Comparison of cyclic loading elongation.

Specimen Number	Elongation at Break (%)	Tensile Ultimate Strength Elongation (%)	Maximum Elongation (%)	Number of Cycles
LL-A	4.3	2.4	4.7	39
LL-B	4.2	1.7	4.5	37
LL-C	3.1	1.1	3.7	28
LL-a	3.7	2.2	3.8	31
LL-b	3.5	1.5	3.6	29
LL-c	3.1	0.8	3.4	27
LY-A	2.4	1.6	2.9	26
LY-B	2.9	1.1	2.9	25
LY-C	3.1	0.6	2.9	26
LY-a	2.2	1.1	2.3	20
LY-b	2.2	0.4	2.3	19
LY-c	2.3	0.6	2.3	20

LL—tension–tension cyclic loading system; LY—tension–compression cyclic loading system.

**Table 7 polymers-15-02728-t007:** Rebar hardening parameter calibration.

Constitutive Model Parameters	${\sigma |}_{0}$ (MPa)	*Q*_∞_ (N/mm^2^)	b_iso_	*C*_kin,1_ (N/mm^2^)	$\gamma_{1}$	*C*_kin,2_ (N/mm^2^)	$\gamma_{2}$	*C*_kin,3_ (N/mm^2^)	$\gamma_{3}$
**Rebar**	429	10	1.2	5000	110	6773	116	2854	34

**Table 8 polymers-15-02728-t008:** Finite-element simulation results.

Specimen ID	*σ*_max_ (MPa)	*FE/Test*	*σ*_max,t_ (MPa)	*FE/Test*	Hysteresis Envelope Area (MPa)	*FE/Test*	*N_B,S_*	*N_B,T_*
A	611.6	0.981	535.93	1.071	218.98	1.114	10	10
B	545.4	0.933	517.52	1.146	143.14	1.130	6	7
C	442.7	0.851	471.84	1.070	103.02	1.145	5	5
D	438.3	0.870	501.25	1.123	90.40	1.205	3	3
E	426.3	0.954	478.82	1.122	105.94	1.274	3	3
F	551.7	0.947	494.38	1.161	83.65	1.146	5	7

*N_B,S_*—the number of cycles when the finite-element buckling phenomenon occurs; *N_B,T_*—the number of cycles when the test buckling phenomenon occurs.

**Table 9 polymers-15-02728-t009:** Values of the parameters considered in the expanded parametric study.

Parameter	Values
Winding layers, *L*	1, 3, 5
Winding angle, *J*	30°, 45°, 60°
Rebar-position eccentricity, *e*/*R*	0.15, 0.22, 0.3

## Data Availability

Data sharing is not applicable to this article.

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
