# Peer review of "Research on Hysteretic Behavior of FRP-Confined Concrete Core-Encased Rebar"

_polymers, 2023, doi:10.3390/polym15122728_

Round 1

Reviewer 1 Report

The manuscript is well written and has good values, but needs some revision before consideration.

1. provide the motivation for the research to undertake this work.

2. What is the life of rubber, when compared to the concrete?

3. How the length of life of the said concrete is improved?

4. Why non-linear stress- strain relationship is obtained for this materials, provide proper explanation.

5. Improve the mechanism of fracture.

6. if possible, provide some micrographs.

7. Improve the conclusions by incorporating future work suggestions.

8. Provide some more recent references.

9. more explanation needed for Fig. 10.

Minor Editing is needed.

Reviewer 2 Report

The authors have experimentally and numerically investigated the performance of the hysteretic behavior of FCCC-R specimens by designing six groups of FCCC-R specimens and four groups of ordinary rebar specimens with different rebar slenderness ratios and confined diameters along with finite element analysis. While the topic is interesting and the model has some merit, the overall manuscript does need additional work to the level of publication in the Journal of Polymers in the opinion of this reviewer. The conclusions drawn in the current paper do not show novel findings compared with previous works. Therefore, significant revisions to the current version of the manuscript are required. Please see my comments below:

- The innovation of the research is not clear. The findings in this research are not new and do not add to the current literature. The introduction lists some relevant research but fails to present a scientific review. Please consider to rewrite and clarify the motivation, objectives, and significance of this study.

- Abstract gives information on the main feature of the performed study, but some more details about the obtained results should be added. The abstract should give an overall summary of the problem statement, proposed methodology, results and conclusion. Please revise the abstract with less specific details about the experimental species.

-Authors must clarify necessity of this study. Aims and scope should be presented in the last part of introduction. (Research significance in presented, but it is not covered aims and scope of the research). The objective statements are rather vague and lacks projected outcomes or how the paper will assist practitioners.

- The literature review merely lists the published work and does not present the major findings from each effort and how it ties to the presented research. The introduction does not identify the knowledge gap that the authors are trying to address with their research.

- In order to better evaluate the advantage of using the system, comparing test results with code estimated capacities is recommended. I would suggest comparing the ultimate bending for the tested specimens with estimated code design values.

- The conclusion needs to be refined; it looks like a discussion. Design recommendations and feasibility need to be discussed further.

- The language of the paper must be checked, since there are a lot of misprints to clean. A substantial internal proof-reading must be developed

- Please add a new section for more discussion on the capacity, such as strength and deformation capacity. A simple comparison between the specimens is not enough, to explain how to evaluate the capacity based on existing knowledge or propose your assessment method is of interest to the readers of the journal.

- The numerical model is not valid unless significant number of validated computational models are considered. How did you validate the accuracy of the model? The presented FE model was not validated with any experimental results. I suggest modeling the tested specimens and compare results with corresponding FE models results for validation.

- The Finite element models for concrete and other materials require a large amount of information. The authors depended in their simulations on several perimeters that some of them obtained from past research and others assumed by the authors. Due to this lack of information, the results from the finite element models presented in the paper is a bit questionable. In general, the manuscript still requires additional revisions in the evaluation of the models presented.

The language of the paper must be checked, since there are a lot of misprints to clean. A substantial internal proof-reading must be developed

Reviewer 3 Report

The Manuscript entitled "Research on Hysteretic Behavior of FRP-confined concrete core-encased rebar" is well designed and described research. The Authors paid attention to all the details describing this research. However there is lack of comparison of the results obtained in this study with other in the fields. This issue has to be revised before publishing.

Other minor issue is the fact that the Authors used sets of references (e.g. "[4-8]"). It has to be explained why all of these works are necessary for this work. 

Round 2

Reviewer 1 Report

The revision made in the manuscript is satisfactory

-

Author Response

Dear Reviewer,

Thank you for your comment. we have made some modifications to the English of this article to make it more fluent and have a higher quality of written statements. Through modifications, we believe that the language statements in the current article are more concise than before.

Thank you and best regards.

Sincerely yours,

Lujingzhou and MouTong

Reviewer 2 Report

The author's addressed reviewer’s comments and as a result the reviewer suggests the paper to be considered for publication

The authors have made a considerable effort to revise the manuscript and the main comments about the scientific approach have been successfully addressed. The paper has been significantly improved however, there are several grammatical mistakes throughout the paper and English requires substantial editing to correct these mistakes and improve the quality of the written presentation using appropriate scientific English language. Also, there are many parts that need to be written concisely and revisions and refinements are required throughout the paper. This is important to ensure that the methodology and the findings of this work are successfully conveyed. 

Author Response

Dear Reviewer,

According to your request, we have made some modifications to the English of this article to make it more fluent and have a higher quality of written statements. Through modifications, we believe that the language statements in the current article are more concise than before.

Thank you and best regards.

Sincerely yours,

Lujingzhou and MouTong